# Green Supply Chains and Digital Supply Chains: Identifying Overlapping Areas

## Viviana D'Angelo * and Valeria Belvedere

Department of Economics and Business Management Science, Catholic University of the Sacred Heart,
20123 Milan, Italy; valeria.belvedere@unicatt.it
* Correspondence: viviana.dangelo@unicatt.it

**Abstract:** This article explores the overlapping between green supply chains and digital supply chains through a bibliometric analysis of the two scientific domains. Using articles' bibliographic data, we conducted a bibliometric analysis of the literature on green supply chains and digital supply chains to examine the intellectual structure of these research domains. By analyzing 131 studies belonging to five different clusters where digital supply chains and green supply chains overlap, our results reveal different overlapping intensity in the different clusters identified. These results reveal also grey areas in the academic research on green and digital supply chains and they may inspire further research explorations, such as addressing whether and how this approach could produce benefits for companies in terms of environmental and operational performance.

**Keywords:** green supply chains; Industry 4.0; digital technologies; bibliometric analysis; research area

## 1. Introduction

Supply chains are highly complex networks connecting suppliers, producers, and clients to ensure the flow of raw materials, intermediate goods, and, lastly, final products to customers. They can be considered the fundamental throughfares involved in creating a product and delivering it to the customer. To use a metaphor employed by Azevedo and colleagues [1], supply chains are like the arteries and veins that transport blood to and from all the cells in the human body, where blood is a metaphor for raw materials and semi-finished and finished products, and the human body is the globe. In the context of the general definition of supply chains, a number of sub-definitions have emerged in response to the latest impactful trends affecting our time—i.e., green supply chains and digital supply chains. Green supply chains (or GSCs) can be defined as production and logistics networks which adopt environmentally friendly practices [2]. The popularity of green supply chains has been amplified due to a twofold reason: growing concern for environmental issues [3] and the positive benefits that can arise in terms of process efficiency and wasting fewer resources when green practices are applied to supply chains and operations [4,5]. Similarly, digital supply chains are production and logistics network based on information systems and innovative technologies that strengthen the integration and agility of a supply chain, thus improving the customer service and sustainable performance of an organization [6]. They have been driven by the digital revolution and the spread of the digital technologies included in the Industry 4.0 paradigm [7], as the adoption of new technologies is one of the main drivers for the performance of the value chains [8,9].

The noteworthy effects that digital technologies have had on efficiency and process improvement have made them one of the key enablers of green supply chains [10], as they are beneficial for operations, reduce the consumption of materials and energy, and shorten operational processes by way of the following [11]: lower demand forecasting errors and, consequently, better inventory control; customized production systems at the cost of mass production; increased production flexibility; shorter lead times and better

capacity utilization; lower supply and time risk; better real-time inventory control; and better coordination between nodes.

Although digital technologies are currently a key enabler of the efficiency required by green supply chains in order to be "green", the two concepts have different roots. The boom of digital technologies within supply chains has contributed to the blurring of boundaries between the two concepts [12], and yet, despite leading to the same result of more efficient output, they belong to different contexts and have different ultimate goals. One has the grandiose, demanding goal of transforming the global economy to benefit all peoples, communities, and the planet by reducing the environmental footprint of business activities, while the other is nurtured by scientific and technological advancements driven by basic and applied research seeking to advance technological trajectories that fuel economic growth. From this perspective, green supply chains and digital supply chains seem to be driven by conflicting forces, and this is the reason the extant literature has explored the two phenomena [13,14]. However, as the boundaries between the two concepts and their research domains have never been clearly defined and, given the lack of clarity regarding the extent to which green supply chains are digital and vice versa (to what extent digital supply chains are green), it is of paramount importance to improve our understanding of how digital technologies converge with green supply chain management and to what extent these concepts are intertwined. To accomplish this purpose, the facets of the literature that lie at the intersection between digital technologies and green supply chains must be dissected.

Previous research has explored the relation between green supply chains and digital technologies by adopting a systematic literature review approach, finding a relationship between digital technologies and environmental strategies. Some studies have focused only on specific digital technologies, i.e., information systems [15], big data and IoT technologies [16–18], or blockchain [19–21], finding that they constitute a means to resolve the issues related to environmental sustainability and to promote a sustainable competitive advantage. On the other hand, Centobelli, Cerchione, and Esposito [22] have focused on information and communication technologies adopted by logistics service providers to identify gaps in the literature and identify future research directions. Others have investigated the link between Industry 4.0 and sustainable manufacturing activities, including supply chain [23], with the aim of identifying gaps in research and opportunities for field development; other scholars have investigated the link between Industry 4.0 and sustainable logistics, identifying managerial opportunities and current and future research trends [24]; and yet others have explored existing research to develop an integrated framework which combines smart, green, resilient, and lean (SGRL) approaches in the context of manufacturing [25].

Apart from the various notable outcomes of systematic literature reviews in summarizing the key findings of extant research and identifying research gaps to be addressed by future researchers [24], it must be acknowledged that this approach involves some structural weaknesses due to its qualitative nature, particularly the selection of papers to be analyzed and its consequent non-replicability. To date, the literature still lacks a comprehensive analysis of research on green supply chains and digital technologies that considers the holistic dimensions of the two concepts and at the same time accounts for their similar ultimate results despite different ultimate goals. Therefore, it is worth conducting a more comprehensive investigation of these two topics by considering both their overlapping areas and their differences in order to clarify their boundaries, common goals and conflicting aspects (if any) that may hamper their ultimate goals. To achieve this purpose, we have conducted a bibliometric analysis of this research domain in order to investigate, firstly, to what extent the concepts of green supply chains and digital supply chains differ in content, and, secondly, to understand whether digital supply chains actually differ from green supply chains. Bibliometric analysis is a quantitative approach to the analysis of research domains and enable to identify, examine and map the intellectual links in the literature regarding, in this case, green supply chains and digital supply chains [26]. By adopting a scientific approach to review, bibliometric analysis overcome the typical

weaknesses of other types of reviews, which are the difficulties in examining a large volume of bibliographic data and the subjectivity of the evaluation, while bibliometric analysis enables to examine all the publications related to a given topic and to cluster similar paper in different sub-areas [27].

We scrutinize and classify the literature linking digital technologies and green supply chains in order to provide better comprehension of the literature streams that, over time, have paved the way towards a meeting of these two domains. In fact, we contend that our understanding of green supply chains would be better if elements of the literature common to the two scientific domains were revealed. From the bibliographic coupling analysis of our dataset, which included 131 articles, we identified the presence of five different clusters: big data, blockchain technology, the circular economy, transportation, and lean and agile approaches. Compared to previous reviews [16–18] which have adopted different perspective, for example analyzing the relation between specific digital technologies and general sustainable manufacturing practices, our study traces the boundaries of the scientific area of digital and green supply chain identifying also the main focus, research patterns and intellectual structure. Our results confirm previous studies, which have found that digital and green supply chains are overlapping concepts, although we have delved more into this issue showing that the two topics have different degrees of overlapping in different areas: high degree of overlapping for what concern transportation strategy and lean and agile approach, while moderate overlapping for what concerns the circular economy and specific digital technologies, such as big data and block chain.

The remainder of this paper is structured as follows: Section 2 provides an overview of the concepts of green supply chains and digital supply chains; Section 3 describes the methodology; Section 4 provides the results of the bibliometric analysis; Section 5 provides the results discussion, and finally, Section 6 provides the conclusions, contributions and future research directions.

## 2. Background

### 2.1. Green Supply Chains

The integration of environmental efforts and global value chains has been defined as green supply chain management (GSCM), a term which refers to a managerial approach adopted within supply chains to improve the environmental performance of processes and products while ensuring profit, market share and operational performance [28–30]. This shift of supply chains toward more conscious, sustainable and environmentally friendly managerial approaches is a consequence of growing concerns regarding the environmental issues that are threatening our world and our society [31–33]. Green supply chains include activities which mirror those adopted within firms [34], namely reducing emissions, minimizing waste, lowering energy use, using renewable materials, incorporating resource conservation measures to ensure that a product/service can be delivered in an environmentally sustainable manner, decreasing the consumption of hazardous and toxic materials, and undertaking reverse logistics actions. Although the external pressures to shift towards sustainable practices, the environmental impact of the supply chains phases has often been neglected [35], especially for what concern the procurement phase while greater attention is devoted toward the distribution phases. Therefore, it is of utmost importance to identify potential enablers of sustainable practices alongside the whole supply chain.

### 2.2. Digital Technologies and Industry 4.0

One of the most impactful trends which is transforming the industrial systems and consequently the supply chains is the diffusion of Industry 4.0 digital technologies. Industry 4.0 is an industrial paradigm which encompasses the integration of digital technologies in production processes in order to improve the efficiency and responsiveness of manufacturing systems [7] and logistics performance [36]. The nine digital technologies included in the Industry 4.0 paradigm has been identified by Boston Consulting Group [37]: big data analytics, simulation modeling, cloud computing, virtual and augmented reality (VR/AR), the

horizontal and vertical integration of systems, the industrial Internet of Things (IIoT), additive manufacturing technologies (3D printing), autonomous robots/automation, and cyber security systems. Supply chains that adopt such technologies are defined as digital supply chains [38] and are characterized by "smart" processes which improve operations and logistics performance, which in turn translate into improved environmental performance.

Although the adoption of digital technologies has been acknowledged as a "must" for operations management and an essential requisite for green supply chains, the effects that digital technologies have on processes include a number of potential pitfalls, both on the operational and the environmental performance side, leading to undesired contrasting effects which limit operational efficiency. For example, although the use of big data improves demand forecasting and enables more reliable production and delivery planning [39–41], it requires additional costs for collecting, storing, analyzing and securing it [42], thereby reducing the beneficial effect it has on operational performance [43]. Therefore, the adoption of several simultaneous digital technologies requires operational tradeoffs on the part of managers and the alignment of long-term strategic goals, operational tradeoffs, and the various digital technologies [44]. Moreover, other studies have found that adopting single technologies leads to different outcomes compared to adopting multiple technologies whose interaction may be detrimental to the final effect [45]. Other researchers have found that the impact of digital technologies on operational performance is mediated by other factors, such as the level of supply chain integration [46], IoT internal capabilities [47], and whether the company in question is a learning organization [48]. On the other hand, similar considerations hold as concerns the effects that digital technologies have on environmental performance. Such complex relationships between digital technologies and operational performance also have an impact on environmental performance. For example, storing big data may require servers that are typically highly energy-consuming, which consequently decreases the overall beneficial effect. On the other hand, a myopic adoption of digital technologies at supply chain level for efficiency purpose without considering the potential beneficial effect on other aspects, such as the environmental footprint, would constitute a potential loss [49]. Therefore, a lack of effective strategic integration of Industry 4.0 with operational and environmental goals may be detrimental for overall performance. For these reasons, the relation between the adoption of digital technologies within the supply chain and the environmental impact improvement must be better investigated.

## 3. Methodology

To find out the overlapping between green supply chains and digital supply chains, we explored the literature by conducting a bibliometric analysis, which is a quantitative method to explore and analyze large volumes of articles which constitutes the cumulative scientific knowledge of the different research fields [27].

The bibliometric analysis was conducted using the VOSviewer software (version 1.6.18) (where VOS stands for Visualization of Similarities), used to construct and visualize maps from bibliographic data. VOSviewer adopts a unified approach to mapping and clustering the bibliometric networks [50] and creates scientific maps where clusters are identified by different colors. Cluster resolution is based on the resolution parameter. In VOSviewer, this parameter can be adjusted to alter the (optimal) number of clusters derived. In line with previous similar studies which adopted the same approach [51], we set the resolution parameter at 0.75, in contrast to the default setting of 1.0, to have a clearer and more consistent distinction between clusters [51–53]. Our decision was based on the fact that excessively detailed clustering would have result in an overly fragmented network that would not be useful for the purposes of our research.

As measure of similarity to map out the research domain, we used the bibliographic coupling strength, in line with previous research studies [51,54,55]. The bibliographic coupling strength between any two documents is defined as the number of articles the two documents share in their reference lists [56], and it is assumed that articles that share the same references address similar topics. The research domain and the connection between

the articles is represented by the bibliographic coupling map, where the strength of the connection is visually highlighted by the closeness of the nodes and thickness of the links. The more the documents cited by both documents, the higher the bibliographic coupling strength.

In order to construct scientific mapping of the research domains explored in this study, we retrieved articles from the "ISI Web of Science Core Collection" database. We set our search string as TS = (green supply chain* AND (digitization OR digitalization OR Industry 4.0 OR Internet of Things OR cloud computing OR artificial intelligence OR information and communication technolog* OR ICT OR big data OR blockchain OR additive manufacturing)). This search led to an initial result of 336 articles. To further check the consistency of the results and the strength of the search string, we split the single keyword "green supply chain" into "green" AND "supply chain" [57–59]. This second search produced 336 articles, in line with the initial search and thus confirming that the search string strategy was suitable, consistent and comprehensive, as no relevant articles were excluded when the unique term "green supply chain" was adopted (TS = (green AND supply chain* AND (digitization OR digitalization OR Industry 4.0 OR Internet of Things OR cloud computing OR artificial intelligence OR information and communication technolog* OR ICT OR big data OR blockchain OR additive manufacturing)). By filtering the results to select the "Web of Science SSCI Index", we obtained a dataset of 176 articles. By selecting the SSCI database we ensured that we were producing a dataset of only articles published in the best peer-reviewed journals that had impact factors and that referred only to social science domains such as business and management, in line with previous researches [58,60].

Finally, we manually screened the results to detect any articles which might not be consistent with the topic. We excluded 43 non-pertinent documents and two articles which were duplicates, resulting in a final dataset of 131 articles. The reasons for the exclusion were the following: not pertaining to the topic of supply chains but of manufacturing in general; not related to digital technologies; and dealing with technology or technological innovation in general. A synthesis of the search procedure is outlined in Figure 1.

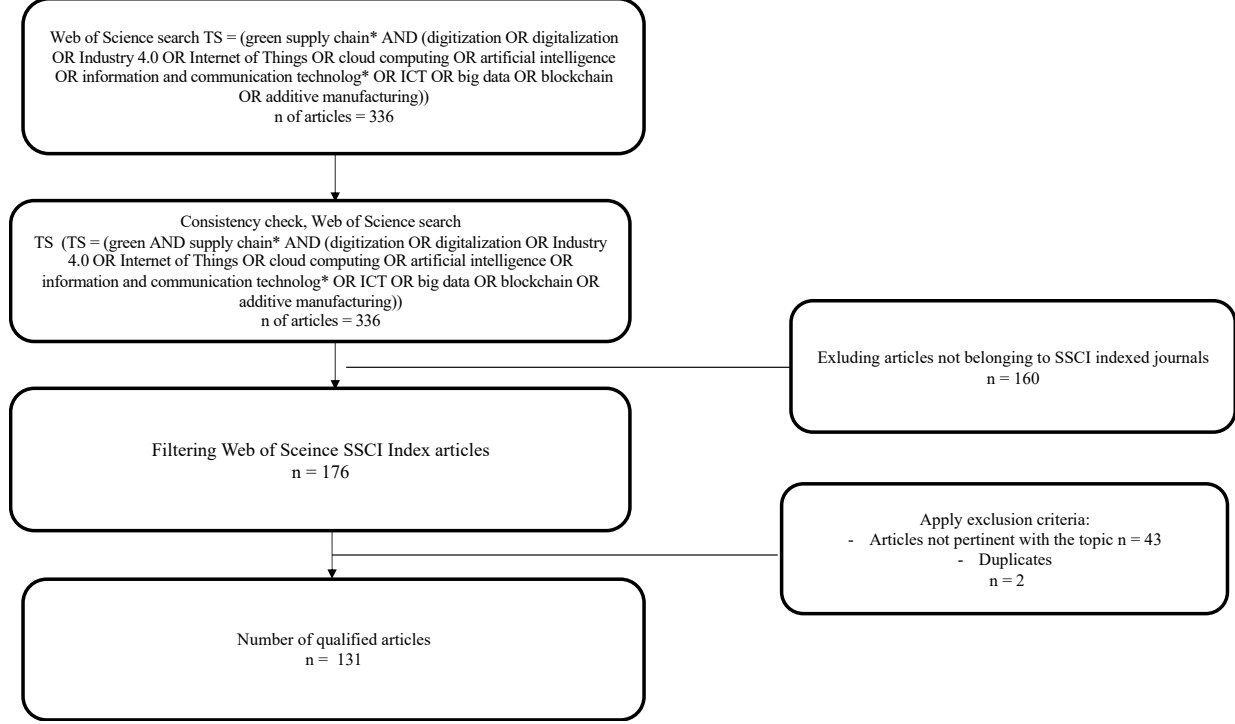

**Figure 1.** The dataset development process, where '*' includes both the singular and plural version of the term.

## 4. Results

### 4.1. Descriptive Statistics

Using general information from the 131 articles retrieved, initial descriptive statistics are provided as a part of our bibliometric analysis. The bar chart in Figure 2 shows the distribution of the articles by topic, according to Web of Science categories. The articles cover a number of research areas, although the largest portion of publications pertains to the field of environmental science and environmental studies. This may indicate the multidisciplinary nature of the theme of green supply chains and digital technologies and it has probably stimulated the cross-fertilization of different fields. Moreover, many papers appear in more than one category, which explains why the sum of articles comprising all research areas is greater than 131. Along with the most relevant areas for business, management, and business and management science, we can also find categories such as environmental science, green sustainable science technology, and environmental studies, confirming the importance of the topics under investigation from an environmental perspective. Other technical and engineering-based areas are industrial engineering, environmental engineering, manufacturing engineering, operations research, and computer science.

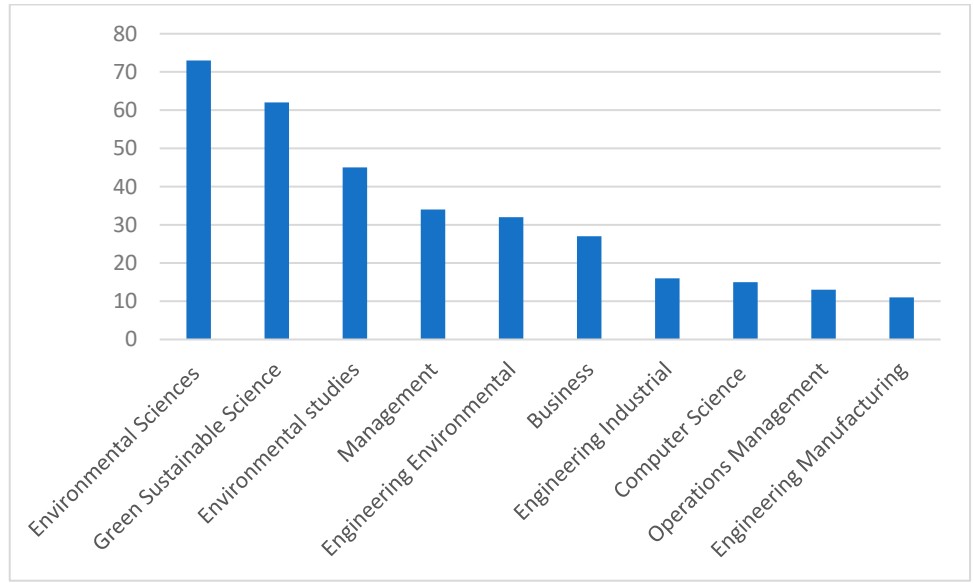

**Figure 2.** Number of publications per category.

The bar chart in Figure 3 reveals the novel nature of the topic, as we can see that the oldest articles were published in 2016, followed by steady growth peaking in 2021. The number of articles for 2022 is incomplete due to the fact that data collection stopped at end of February 2022.

The distribution of the articles in terms of journals shows a low level of concentration, with many different journals contributing to the publication of the 131 articles in our dataset. A deeper look at the most prolific journals enables us to identify the most favorable outlets for the topics under investigation. At the top position we find Sustainability with 24 articles, followed by the Journal of Cleaner Production with 18 articles, Technological Forecasting and Social Change with 8 articles, and Resources Conservation and Recycling with 7 articles. These are followed by a few journals with 4 articles each: Business Strategy and the Environment, the International Journal of Production Research, and the Journal of Enterprise Information Management. Other journals published only 3 articles: Industrial Management & Data Systems, IEEE Access, the International Journal of Environmental Research and Public Health, and the Transportation Research Part E-Logistics and Transportation Review. The rest of the articles are spread across several other journals which published one or two articles. The bar chart in Figure 4 shows the journals that have published more than 4 articles.

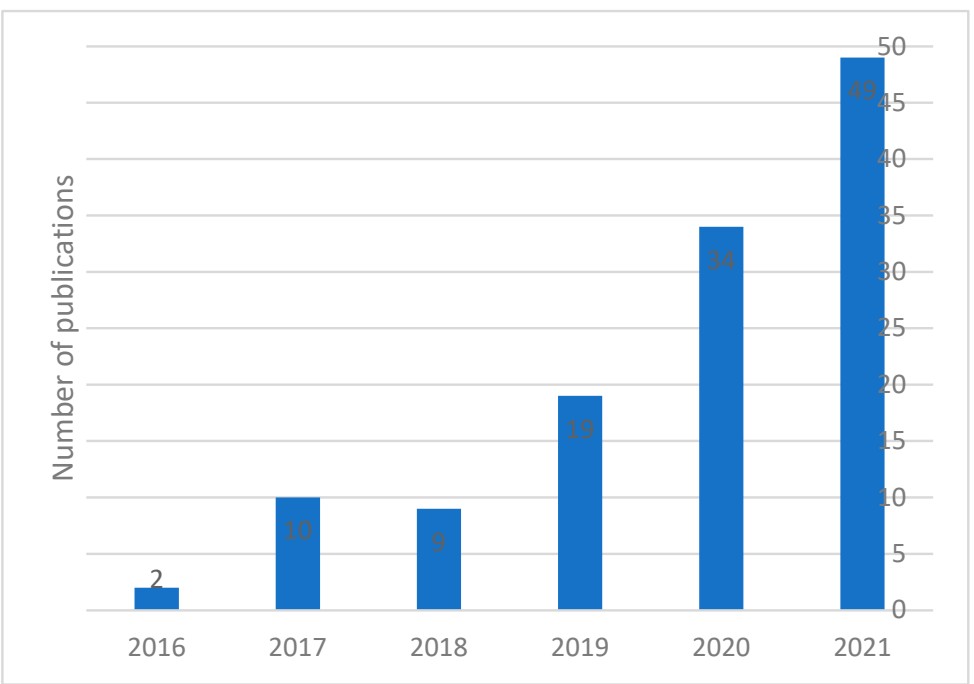

**Figure 3.** Number of publications per year.

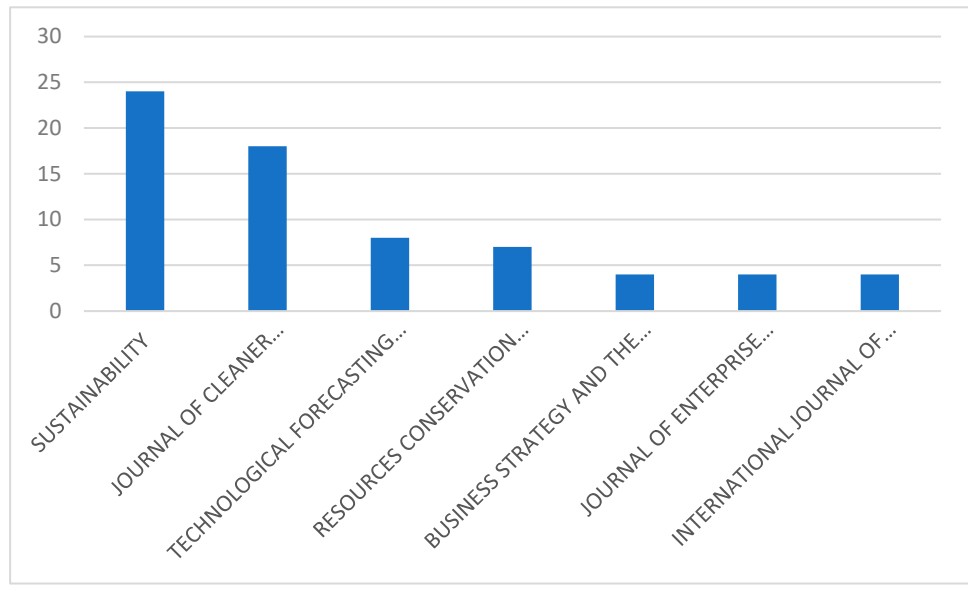

**Figure 4.** Number of publications by journal.

### 4.2. Bibliographic Coupling Map

The scientific map obtained through analysis of the dataset conducted with VOSviewer is reported in Figure 5. Each node represents an article, while each link represents the bibliographic-coupling relationship. The size of the nodes represents the number of citations the article has received over time, revealing the scientific importance of the article, while the distance between nodes represents similarity or difference between articles (as measured by bibliographic coupling). Finally, the color of the nodes identifies the clusters the articles belong to.

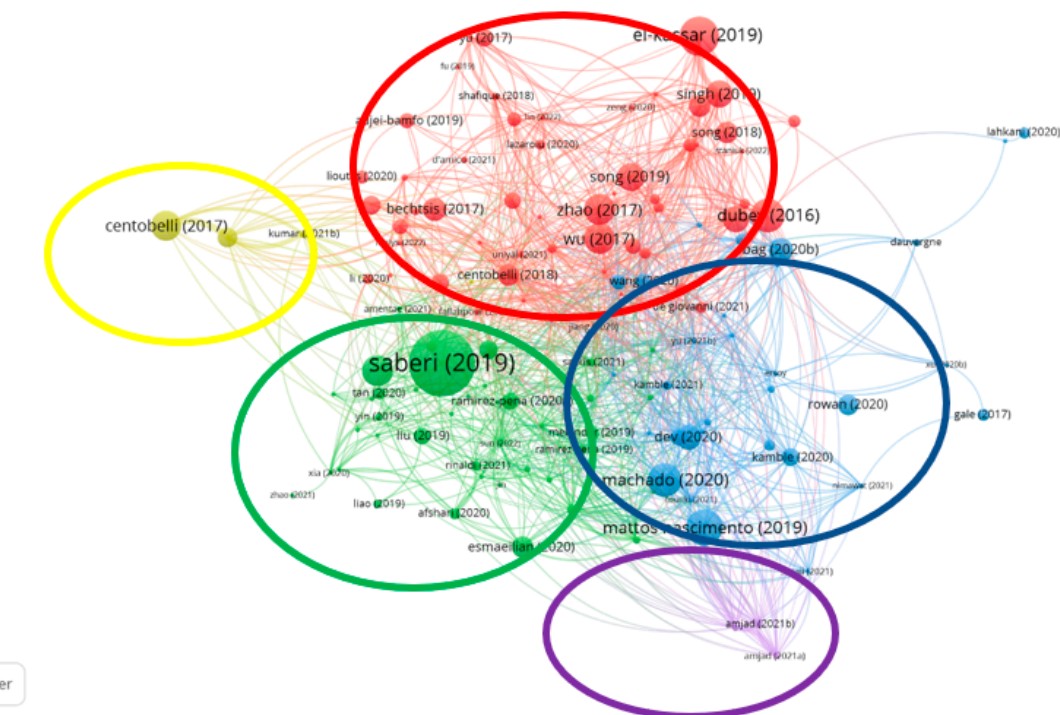

**Figure 5.** Bibliographic coupling map with resolution parameter equal 0.75 (First Autor, Year).

The VOSviewer 1.6.18 software identified five different clusters: big data, blockchain technology, circular economy, transportation, and lean and agile approaches. We conducted content analysis of each cluster to identify the focus of each study and the most frequently recurring topics, theoretical frameworks, and key insights.

### 4.2.1. Cluster 1—Red Cluster—Big Data

Cluster 1, shown in red on the map in Figure 5, includes 54 articles, and it is therefore the largest cluster in the scientific domain. Our analysis shows that this cluster focuses on the role that big data and the application of big data analytics play in supply chains. A closer examination of the papers belonging to this cluster reveals that the role of big data is investigated at various levels: its role in enhancing the three dimensions of performance (organizational, economic, and environmental) in supply chains thanks to its ability to mitigate the risk of supply chain disruption [61]; its role in improving evaluations and decision-making [62–64]; and its role in reducing transaction costs [65]. Furthermore, the role big data plays in supply chain management and procurement is investigated [62,66], particularly in regard to the selection of green suppliers [67,68], the development of an ICT platform for collaboration and sustainable supply chains planning [69], sustainable public procurement [70,71], improving e-procurement [72], and fostering collaboration in supply chains in general [73–75].

Another interesting aspect that arises from this analysis is the use of surveys as the primary methodology to collect data (14 out of 53 articles). A survey involves the collection of information from individuals (through mailed questionnaires, telephone calls, personal interviews, etc.) about themselves or about the social units which they belong to. The reason for the popularity of survey analyses might be the lack of availability of secondary data in the operations and management field [76]. These survey data sets are mostly analyzed using structural equation modelling (SEM), which is a quantitative approach to exploring relationships between latent variables, i.e., abstract concepts not directly observable but measured through constructs defined by observable variables [77]. In this sense, a more in-depth analysis of the constructs used to estimate latent variables in the different articles, relative to the results obtained, would be useful. In fact, given that latent variables are

defined by constructs selected by researchers (even though the previous literature provides constructs which have been validated), it would be interesting to examine these results in light of the different and possibly new constructs used.

### 4.2.2. Cluster 2—Green Cluster—Blockchain Technology

Cluster 2, which is the green one on the scientific map in Figure 5, focuses on the application of blockchain technologies in order to improve logistical efficiency, and it contains 41 articles. Blockchains can be defined as decentralized, distributed, and transparent ledgers in which transactions are recorded in a chronological order [42]. Overall, this cluster supports the view that the implementation of blockchain technology in supply chains improves monitoring, data collection and the exchange of information, thus improving overall efficiency and consequently reducing environmental footprints. A closer analysis reveals that the papers explore different dimensions of blockchain technologies in logistics and supply chains. Some authors focus on inter-organizational barriers to the implementation of blockchain technology, system-related barriers, external barriers, and on developing propositions for implementing blockchains in supply chains [78]. Other authors have identified the most successful applications for blockchains: vendor selection, supplier development, purchasing and materials management, inbound logistics, production and internal operations, eco-design and life cycle assessment (LCA), outbound logistics, waste management, and reverse logistics [79]. Some authors have identified further benefits: reducing the cost of verification and the cost of networking; improving reporting systems; ensuring more efficient transportation and shipment tracking; and reducing the bullwhip effect, which is a supply chain phenomenon in which small fluctuations in demand at the retail level can cause progressively larger fluctuations in demand at the wholesale, distributor, manufacturer and raw material supplier levels. However, some scholars have also identified the "dark side" of blockchain technology, namely its considerable energy footprint due to the computational-intensive nature of current blockchain protocols [80]. Other authors have identified challenges in implementing blockchain technologies, such as the problem of data storage and transmission, implementation costs and risk (i.e., device costs, training costs, operational costs and maintenance costs), and the need to adopt a proper incentive mechanism to push the logistics industry to record and store data (the logistics industry obtains few rewards when logistics companies record data on logistics processes) [81].

An interesting aspect emerging from our analysis is the overlap between this topic and themes emerging from other clusters, such as the circular economy, which is the focus of cluster 3, identified in blue on the map. We might rightfully wonder why the paper by [82], which focuses on the role of blockchain technology in the circular economy, has been assigned to cluster 2 and not to cluster 3. The explanation for this attribution can be found in the algorithm analytical process: the paper shares most of its literature with papers focusing on blockchain technology rather than on the circular economy; therefore, we can deduce that the main focus of this paper is blockchain technology application.

### 4.2.3. Cluster 3—Blue Cluster—Circular Economy

Cluster 3, identified in the bibliographic coupling map as the blue cluster, includes 29 articles, and it focuses on the applications of circular economy practices in supply chains and other collateral topics in the circular economy, i.e., reverse logistics [83–85] and managing the end-of-life process for products [86,87]. Although our analysis focuses on the scientific domain of green supply chains and digital supply chains, and even though the circular economy relates to the domain of green and sustainable practices, circular economy applications being in a cluster is not surprising. In fact, the concept of the circular economy is a more specific topic than green and sustainability practices, as it refers to a production paradigm which aims to close the loop of a production cycle, mimicking living biological systems [88]. On a second level, the subarea indicates interest in the issue of the sustainable performance of supply chains [73], and in the barriers and challenges to the

adoption of circular economy practices in green supply chains, such as a lack of a skilled workforce, a focus on short-term goals and ineffective strategies [89], or a lack of clarity regarding the benefits of digitalization, implementation costs, and a lack of standards and regulations [25,90].

The most popular industrial sector explored by the community is that of the food industry [91,92]. The reason for this might be due to the importance of the issue of food waste within supply chains and the importance of an effective network to efficiently manage food security, food surplus, food loss, and waste [91], in line with the need to slow down the depletion of resources and to comply with green and sustainability standards [93]. On the other hand, some researchers belonging to this cluster have explored the reverse logistics of more complex products, such as vehicles [87] or refrigerators [85], highlighting the hidden burdens that reverse logistics may imply. In fact, recycling complex manufacturing products, for example vehicles at the end of their product life, might not be cost-effective, as most vehicles have many components made of different materials with varying degrees of renewability. This complicates the process of recycling in terms of cost and time, and therefore the inappropriate management of end-of-life products could be detrimental to the environmental and cancel the economic benefits of such actions.

If we observe the cluster from a different perspective, it emerges that one of the most frequently adopted methodologies in the cluster is mathematical modeling, possibly tested through computer simulations. Some articles have used ad hoc mathematical models to fit their research purposes [84,87,94], while other papers have applied established mathematical models such as analytic hierarchy process [89,95], large group decision-making (LGDM) approach [25], and the Fermatean fuzzy CRITIC-COPRAS method [96]. This is an interesting aspect to note because the use of mathematical modeling in the field of operations and logistics is considered outdated: "if we compare contemporary research in operations management (OM) with that conducted in the early 1980s, we notice an increase in the use of empirical data (derived from field observation and taken from industry) to supplement mathematics, modelling, and simulation to develop and test theories" [76].

It is worth mentioning that one paper discussed the obsolescence of standard-setting organizations when defining the "green" standards that regulate and formally assess circular economy activities in green supply chains since the rise of digital technologies and Industry 4.0 [97]. According to the authors, since these technologies enable traceability, trustworthiness, real-time data collection and analysis, they neutralize the role of audit committees, which are outdated, low-tech and costly [97]. Besides its notable contribution to the debate regarding the obsolescence of international and intergovernmental committees, this paper raises noteworthy and important points on the role of digital technologies spanning much broader boundaries than simply the efficiency of operations and supply chains. Unfortunately, however, as the paper concludes, revising deep-rooted routines, established organizations, and important institutional forces may be difficult. A potential solution provided by the article is that of reconciling the strength of digital technologies by decreasing the cost of compliance audits, reducing the time taken to access information, and improving the quality and quantity of this information, thus respecting the role of standard-setting organizations. Unlike the rest of the articles in this entire research domain, this study deals with an issue of great interest for policy makers since the journal which published the article is oriented towards global political challenges and, in particular, the relationship between global political forces and environmental change. This fact shows the broader perspective of economic policy journals that embrace a wider and more diversified view of digital technologies and green supply chains, which could provide an interesting stimulus for future research projects.

### 4.2.4. Cluster 4—Yellow Cluster—Transportation

Only four articles belong to cluster 4, the yellow one on the map in Figure 5. It is rather small compared to the others and it is highly sector specific. Our analysis has shown that it focuses only on the environmental performance of transportation in logistics

and supply chains. The small dimension of this subset, which nonetheless constitutes an independent subfield, is per se an interesting result. It demonstrates that research on this topic, despite being one of the most important groups for logistics and supply chains, is quite marginalized, while also highly specialized. In fact, issues related to transportation are singled out by engineers and mathematicians interested in mathematical modeling and optimization problems, which are typically beyond the scope of management scholars.

### 4.2.5. Cluster 5—Purple Cluster—Lean and Agile Approaches

Cluster 5, the purple one in Figure 5, is the smallest one, containing only three articles. Upon closer examination, it became clear that this cluster focuses on the role of lean and agile approaches in green manufacturing systems, which is a segment of the whole supply chain. This also explains the absence of articles focusing on lean approaches in other clusters. Similar to the issues that emerged in Cluster 4, the limited numbers of papers in Cluster 5 could be explained by the fact that such "traditional" topics as lean and agile methods target academic journals other than those dedicated to management or, perhaps, they refer to lean and agile methods without highlighting the connections between these concepts and green-related themes. This fact constitutes an important issue for research on operations and logistics management, which is threatened by the risk of becoming a marginal topic in the eyes of top management, even though this dimension of companies is simultaneously fundamental and completely disconnected from other parts of companies.

### 5. Discussion

A summary of the results of our cluster analysis is reported in Table 1.

**Table 1.** Cluster summary.

| Cluster | No. of Articles | Methodology |
|---|---|---|
| C1—Red Cluster—Big Data | 54 | survey |
| C2—Green Cluster—Blockchain Technology | 41 | mathematical modeling |
| C3—Blue Cluster—Circular Economy | 29 | mathematical modeling |
| C4—Yellow Cluster—Transportation | 4 | mathematical modeling |
| C5—Purple Cluster—Lean and Agile | 3 | case studies |

The results of our bibliometric analysis clarify whether and to what extent the topics of green supply chains and digital supply chains are intertwined. Cluster 1 (big data) and Cluster 2 (blockchain technology) focus on technological dimensions and advantages prompted by the novelty of these technologies, particularly big data and blockchains. In these two research domains, scholars are mostly interested in understanding the potential strength of these technologies and related opportunities. Therefore, based on our content analysis of these research articles, we can conclude that the intersection between the topics "green" and "digital" in Cluster 1 and in Cluster 2 is low. However, such a strong focus on the technological dimension can have a negative effect on the advancement of these research domains. In fact, an excessive focus on technological aspects of supply chains may divert attention from the managerial aspects of supply chain management, exacerbating long-term distance from a company's strategic view.

On the other hand, Cluster 3 (the circular economy) focuses on identifying possible circular economy business models and business opportunities related to closed-loop productions, and how to implement such business models to take advantages of potential opportunities. Therefore, we can conclude that the intersection between the topics "green" and "digital" in Cluster 3 is low. The topic of the circular economy has moderate relevance to supply chains per se, while it deals better with business model innovations which supply chains are part of.

Cluster 4 (transportation) and Cluster 5 (lean and agile approaches) can be defined as research areas that deal with topics traditionally related to logistics (Cluster 4) and

operations (Cluster 5). In this sense, "transportation" and "lean and agile approaches" clusters echo themes related to efficiency and logistics, and to the problem of managing flows while minimizing costs. Digital technologies are of great importance as concerns the green performance of transportation and logistics; therefore, the topic of sustainability shows strong relevance in Cluster 4. Similarly, as concerns lean and agile approaches in supply chains as well, the adoption of digital technologies is an important enabling factor to promote lean practices. For example, wearable technologies like smart glasses can be used in warehouses to increase the speed of operations and efficiency. Therefore, we can conclude that the intersection between the topics "green" and "digital" is strong both in Cluster 4 and Cluster 5. However, the limited number of papers comprised in these clusters reflects a peculiar yet interesting trait of this research domain, since scholars typically analyze the topics of transportation and lean and agile operations as single phases of the supply chain, rather than considering them within the context of the whole supply chain. The synthesis of the overlapping degree of the topics is summarized in Figure 6.

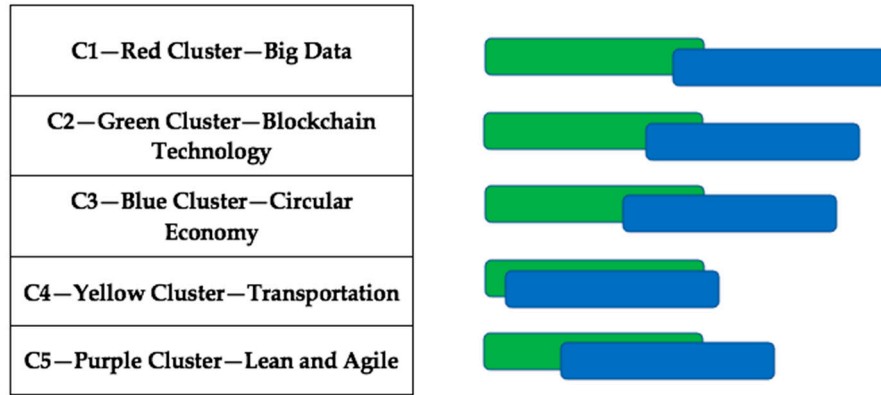

**Figure 6.** Degree of overlapping of "green" and "digital".

Focusing on transportation, we can observe that in the recent years a number of studies have tried to shed light on managerial practices that can lead to superior environmental performance in logistics and transportation [98–100], while others have examined the role of technology in enhancing transportation performance [101,102].

Similarly, many recent papers have addressed the topic of lean management as an enabler of digital and Industry 4.0 technologies [103–105], demonstrating that in most cases a synergy between the two can be observed. Furthermore, lean management is frequently mentioned as a paradigm that, being committed to waste reduction, can be fruitfully adopted in companies striving to improve their environmental performance [106–109]. However, our bibliometric analysis demonstrates that technological and environmental perspectives are seldom observed jointly in papers on transportation and lean management. This evidence points to a research opportunity, since the robust connection between these constructs is quite evident. Moreover, it must be noted that, while the literature on supply chain management reveals greater interest in coordination and cooperation among players in the pipeline, topics such as transportation and production (which lean management refers to) are more frequently addressed using an internal perspective. This feature can influence the way in which scholars position their papers on transportation and lean management within the wider field of supply chain management. Thus, we conclude that studies in these two areas should better pinpoint the relevance of supply chain relationships in terms of improving environmental performance. For instance, lean companies could benefit from closer coordination with suppliers not only to improve their operational performance but their environmental performance as well. Similarly, $CO_2$ emissions in transportation could be reduced sharply if collaborative practices with clients were adopted.

Another interesting aspect that emerges from this analysis is the lack of a cluster focused on supply and procurement, although they are important phases in supply chains. A few articles can be found in Cluster 1. Some were focused on public procurement [70,71],

while others focused on the impact big data has in e-procurement [72,110]. This limited interest in the role those digital technologies play in the "greening" of the procurement phase of supply chains tells us how researchers perceive digital technologies, namely that they strongly affect the procurement phase but do not affect environmental performance. We invite future researchers to shed more light on this relationship, exploring whether there is a hidden environmental impact of procurement and how digital technologies could possibly address it.

## 6. Conclusions

The contributions of this study are manifold. First, our study provides a more comprehensive assessment of the intertwining of green supply chains and digital technologies. Our results confirm the findings from previous studies [15,16,19], which found digital technologies being one of the key enablers of green supply chains. However, our results go beyond those findings, showing that the intersection between digital and green supply chain has different level of overlapping in different context identified by different research subfield. We found moderate overlapping in the domains of big data, blockchain, and circular economy. These results point out the significant room for improving the synergic combination between green and digital. In fact, big data and blockchain technologies should be further leveraged to address "green" and environmental performance improvement, an aspect which is typically left aside as an incidental, although positively welcomed, effect. Similarly, the circular economy paradigm could benefit more from the use of digital technologies. So far, the applications of digital technologies in advancing the circular economy practices are not being adequately explored. This paves the way to unlock the potential effect of combining digital technologies and circular economies, finding new interdependencies which would stimulate the diffusion of circular economy practices. On the other hand, we found significant overlapping between green and digital in transportation. This highlights the importance of digital technologies to solve the issues concerning the transportation strategies and route optimization. For example, the predictive algorithms, the IoT technologies and the use of big data enable a constant and accurate forecast of the optimal route to reduce the fuel consumptions. Also, in the lean and agile manufacturing field the overlapping between the two concepts is significant. This comes from the need to adopt digital technologies to foster lean manufacturing approach, for example adopting IoT based smart bin to improve the flow alongside the entire processes, or using advanced predictive algorithm based on big data to refine the demand forecasting and to eliminate the inventories, and finally adopting sophisticated augmented reality tools to support the quality control.

Second, our study considers the full spectrum of Industry 4.0 digital technologies applied to all supply chains phases, aspects, and dimensions, while our research compared to previous bibliometric analyses, which analyzed the relation of environmental efforts and specific digital technologies [4,111,112].

Third, the methodology employed enabled us to analyze the entire research domain adopting a common scientific assessment approach overcoming the main weakness of systematic literature reviews. Our study is also relevant for management, in particular for those in leading positions, because it highlights the need to re-align long-term strategic goals and operational performance dimension. In addition, our results are relevant for policymakers who are looking for ways to face the urgent environmental challenges affecting our era, i.e., resource degradation and climate change.

Our results contribute to understanding of the impact that Industry 4.0 digital technologies have on green supply chains by providing a replicable assessment of scholarly knowledge regarding these two topics, while identifying weaknesses and the ways future researchers could address certain grey areas. Firstly, supply chain management (as well as operations management) is typically considered a marginal area of management studies, whereas it is of great interest for technicians and engineering researchers. Given the importance of top management support and, in general, of managerial perception of

digital technologies in stimulating their adoption within companies [65], it is of utmost importance to include a study of operations and supply chains from the perspective of management as well. We invite management researchers to shift their focus to supply chain management and operations management and to include these themes from the perspective of top management. Moreover, we invite future researchers to broaden their analysis of the digital technologies which enable green supply chains by exploring the impact of technologies other than big data and blockchains.

Other important advice for future researchers emerges from the limitations of our study. First, our dataset was retrieved from Web of Science SSCI, but future researchers could explore different databases. Second, we used bibliographic coupling as the measure of similarity, assuming that documents that share parts of their bibliographies are similar, but future researchers could develop other and more sophisticated similarity measures, e.g., based on AI, to refine research results. Finally, the content analysis conducted by VOSviewer 1.6.18 software to define the clusters was applied to the title, abstract, and keywords, excluding articles not containing the keyword in their title or abstract, whereas future researchers could develop new algorithms to process bibliographic data accessing the entire research database rather that only the title, abstract and keywords.

Our results show an overall moderate level of overlap between digital technologies and green supply chains. This is in line with the different ultimate goals of each topic. Industry 4.0 digital technologies are based on connectivity, integration and the digitization of systems, and they look to the future of technological advancements, where the boundaries between the digital and physical worlds will become blurred, as will integration between human and machine agents, materials, products, production systems and processes. Green supply chains improve our environmental footprint, increase resource conservation and promote social goals, even though they may, if necessary, sacrifice technological advancement. All in all, it can be concluded that digital supply chains and green supply chains operate in a parallel manner, but sometimes their tracks converge, and society will only be able take maximum advantage of these paradigms when they are considered a single entity.

**Author Contributions:** Conceptualization, V.D. and V.B.; methodology, V.D.; software, V.D.; formal analysis, V.D.; writing—original draft preparation, V.D.; writing—review and editing, V.B.; supervision, V.B. All authors have read and agreed to the published version of the manuscript.

**Funding:** This research received no external funding.

**Institutional Review Board Statement:** Not applicable.

**Informed Consent Statement:** Not applicable.

**Data Availability Statement:** Data available on request.

**Conflicts of Interest:** The authors declare no conflict of interest.

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
