# Peer review of "Green Supply Chains and Digital Supply Chains: Identifying Overlapping Areas"

_sustainability, doi:10.3390/su15129828_

Round 1
Reviewer 1 Report
- Paragraphs are too long and unreadable. It is necessary split them accordingly.
- The topic seeks to catch the reader's attention but is not efficient in the attempt. Indicate de proposal of the article clearly in the title.
- the context of the study is not explored in the abstract, the results are general and vague in describing what was obtained.
- Section 2 seeks to establish the idea between Green Supply Chains and Industry 4.0 but the correct was explored Digital Transformation. The section should be explored in a separate form.
- the methodology explores the idea of bibliometric analysis but I think is necessary to justify choices. Why did you use those terms?
- Figure 1 is ineligible, use the same font of the text.
- The first paragraph o item 4.2 is methodology.
- Use the same font size over the text.
- Rethink manuscript construction. What does your paper go beyond all other literature reviews?
no comments at all
Author Response
Reviewer #1
Comments to the Author
Reviewer 1_1
- Paragraphs are too long and unreadable. It is necessary split them accordingly.
Answer to Reviewer 1_1
We are grateful to Reviewer 1 for this comment. We have split the paragraph 2 "Green Supply Chains and Industry 4.0” in two subparagraphs, and we have split the paragraph “Discussion and Conclusions” in two different paragraphs. We hope that this amendment would improve the readability of the manuscript.
Reviewer 1_2
- The topic seeks to catch the reader's attention but is not efficient in the attempt. Indicate de proposal of the article clearly in the title.
Answer to Reviewer 1_2
We are grateful to Reviewer 1 for this insightful suggestion. We have modified the article title to provide a more immediate sense of the purpose, and now the paper titles “Green Supply Chains and Digital Supply Chains: identifying overlapping areas through a bibliometric analysis”.
Reviewer 1_3
- The context of the study is not explored in the abstract, the results are general and vague in describing what was obtained.
Answer to Reviewer 1_3
We thank Reviewer 1 for pointing this. We have better clarified in the abstract the context of our research, the methodology and the results obtained.
Reviewer 1_4
- Section 2 seeks to establish the idea between Green Supply Chains and Industry 4.0 but the correct was explored Digital Transformation. The section should be explored in a separate form.
Answer to Reviewer 1_4
We are thankful to Reviewer 1 for this comment. We have now better explained in section 2.2 the link between Digital Supply Chains, Digital Technologies and Industry 4.0.
Reviewer 1_5
- the methodology explores the idea of bibliometric analysis but I think is necessary to justify choices. Why did you use those terms?
Answer to Reviewer 1_5
We are grateful to Reviewer 1 for raising this point. We have now better explained in the Introduction and in the Methodology sections why a valid methodology to address our research question is the bibliometric analysis.
Reviewer 1_6
- Figure 1 is ineligible, use the same font of the text.
Answer to Reviewer 1_6
We are thankful to Reviewer 1 for this comment. We have modified the Figure 1 to improve the clarity and the readability of the figure and now the size of the font is homogeneous.
Reviewer 1_7
- The first paragraph o item 4.2 is methodology.
Answer to Reviewer 1_7
We are grateful to Reviewer 1 for pointing this. We have now moved the first paragraph of the section 4.2 in the Methodology section.
Reviewer 1_8
- Use the same font size over the text.
Answer to Reviewer 1_8
We thank Reviewer 1 for this comment. We have now aligned the font size over the whole manuscript.
Reviewer 1_9
- Rethink manuscript construction. What does your paper go beyond all other literature reviews?
Answer to Reviewer 1_9
We are thankful to Reviewer 1 for this comment. We have stressed in the Introduction and Conclusions sections the theoretical contribution and how our results add compared to the previous articles which have conducted other types of literature reviews.
All the amendments implemented in the revised version of the paper have been highlighted in red. We hope that with this revised version we have addressed all the insightful concerns you have raised, and that our work could be now considered for publication in Sustainability.
Best regards,
The Authors

Reviewer 2 Report
The paper presents a systematic literature review on green supply chains and digital supply chains. It is a topic of interest to the researchers in the related areas but the paper needs very significant improvement before it is considered for publication. My detailed comments are as follows.
1. Originality: This article explores the relationship between green supply chains and Industry 4.0 digital technologies, focusing on the extent to which these two topics relate and overlap, which is new and significant, but the significance of this paper is not expounded sufficiently. The author needs to highlight this paper's innovative contributions. Specially, please specify the contribution of this paper in Introduction and Conclusion parts and explain how the paper fill up the research gap.
2. Relationship to Literature:The author only shows the concepts of green supply chain and digital supply chain without in-depth analysis, which is difficult to reflect the research value of this paper. It is suggested that the author add more discussions about green supply chain and digital supply chain, and point out the existing research gap to provide support for the research significance of this paper.
3. Research Framework: The framework for the study of this manuscript is imperfect. In lines 122-125, “The remainder of this paper is structured as follows: section 2 provides an overview of the concepts of green supply chains; section 3 provides an overview of the concepts of Industry 4.0 and digital technologies; section 4 provides the results of our bibliometric analysis; and, finally, section 5 provides a discussion, conclusions, and future research directions”. This statement is inconsistent with the research framework of this paper, it is suggested that the author reframe the research framework and contents properly.
4. Methodology: The paper uses bibliometric analysis to map out the research domain. However, some of the charts are not up to standard, such as “Figure 3 Number of publications per year”, the ordinate name in Figure 3 should be given explicitly. In lines 245-257, the number of publications in each journal should be shown more visually in chart form.
5. Conclusions: The manuscript lacks a conclusion, the author mixed the research results and conclusions together, which blurred the research significance and failed to highlight the main contribution of the paper. It is suggested that the author add the research conclusion and discussion, thoroughly analyze the degree of overlapping of “green” and “digital in big data, blockchain technology, the circular economy, transportation, and lean and agile approaches.
There are some formatting errors in this manuscript. In lines 412-423, and in lines 547-556, font size and paragraph spacing do not match the rest of the content.
Author Response
Reviewer #2
Comments to the Author
Reviewer 2_1
The paper presents a systematic literature review on green supply chains and digital supply chains. It is a topic of interest to the researchers in the related areas but the paper needs very significant improvement before it is considered for publication. My detailed comments are as follows.
Answer to Reviewer 2_1
We are grateful to Reviewer#2 for the timely and constructive comments provided to us.
Reviewer 2_2
Originality: This article explores the relationship between green supply chains and Industry 4.0 digital technologies, focusing on the extent to which these two topics relate and overlap, which is new and significant, but the significance of this paper is not expounded sufficiently. The author needs to highlight this paper's innovative contributions. Specially, please specify the contribution of this paper in Introduction and Conclusion parts and explain how the paper fill up the research gap.
Answer to Reviewer 2_2
We are thankful to Reviewer 2 for raising this point. We have now highlighted the contribution of this paper in the Introduction and Conclusions sections and how our results add to the findings of previous studies.
Reviewer 2_3
Relationship to Literature:The author only shows the concepts of green supply chain and digital supply chain without in-depth analysis, which is difficult to reflect the research value of this paper. It is suggested that the author add more discussions about green supply chain and digital supply chain, and point out the existing research gap to provide support for the research significance of this paper.
Answer to Reviewer 2_3
We are thankful to Reviewer 2 for this comment. We have now deepen the concepts of green supply chain and digital supply chain in section 2 highlighting the main concerns that they raise, especially in relation to the strategic purpose of each, and consequently highlighting the importance of clarifying their boundaries.
Reviewer 2_4
Research Framework: The framework for the study of this manuscript is imperfect. In lines 122-125, “The remainder of this paper is structured as follows: section 2 provides an overview of the concepts of green supply chains; section 3 provides an overview of the concepts of Industry 4.0 and digital technologies; section 4 provides the results of our bibliometric analysis; and, finally, section 5 provides a discussion, conclusions, and future research directions”. This statement is inconsistent with the research framework of this paper, it is suggested that the author reframe the research framework and contents properly.
Answer to Reviewer 2_4
We are grateful to reviewer 2 for pointing this. We have amended papers’ summary provided at the end of the Introduction section and now the content of the statement is aligned with the right paragraph numbers.
Reviewer 2_5
Methodology: The paper uses bibliometric analysis to map out the research domain. However, some of the charts are not up to standard, such as “Figure 3 Number of publications per year”, the ordinate name in Figure 3 should be given explicitly. In lines 245-257, the number of publications in each journal should be shown more visually in chart form.
Answer to Reviewer 2_5
We thank Reviewer 2 for this suggestion. We have included in the Figure 3 the description of the vertical axis.
Reviewer 2_6
Conclusions: The manuscript lacks a conclusion, the author mixed the research results and conclusions together, which blurred the research significance and failed to highlight the main contribution of the paper. It is suggested that the author add the research conclusion and discussion, thoroughly analyze the degree of overlapping of “green” and “digital in big data, blockchain technology, the circular economy, transportation, and lean and agile approaches.
Answer to Reviewer 2_6
We thank Reviewer 2 for this comment. We have now enriched the Conclusion section highlighting the theoretical contribution of our study and how our results relate to the previous literature. In addition, in order to highlight the conclusions, we have split the Discussion and the Conclusions sections in two different paragraphs.
Reviewer 2_7
Comments on the Quality of English Language
There are some formatting errors in this manuscript. In lines 412-423, and in lines 547-556, font size and paragraph spacing do not match the rest of the content.
Answer to Reviewer 2_7
We thank Reviewer 2 for this comment. We have now aligned the font size and the paragraph space throughout the whole manuscript.
All the amendments implemented in the revised version of the paper have been highlighted in red. We hope that with this revised version we have addressed all the insightful concerns you have raised, and that our work could be now considered for publication in Sustainability.
Best regards,
The Authors
Reviewer 3 Report
The article is devoted to the topical topic of social development of two areas of green chains and digital chains. The methodology of the article is clear, the authors show and justify the marginality of gray zones in green chains. However, the general change of the existing economic system to the industry 4.0 paradigm, although it brings significant changes, we cannot yet draw a clear line between green and digital. Nevertheless, the authors have made a more or less successful attempt to cluster the differences between green and digital, which is likely to be useful for further research is already possible on specific examples within a large logistics organization.
Author Response
Reviewer #3
Comments to the Author
Reviewer 3_1
The article is devoted to the topical topic of social development of two areas of green chains and digital chains. The methodology of the article is clear, the authors show and justify the marginality of gray zones in green chains.
Answer to Reviewer 3_1
We are grateful to Reviewer 2 for acknowledging the relevance of the topic and the methodology.
Reviewer 3_2
However, the general change of the existing economic system to the industry 4.0 paradigm, although it brings significant changes, we cannot yet draw a clear line between green and digital. Nevertheless, the authors have made a more or less successful attempt to cluster the differences between green and digital, which is likely to be useful for further research is already possible on specific examples within a large logistics organization.
Answer to Reviewer 3_2
We are thankful to Reviewer 2 for this feedback.
Round 2
Reviewer 1 Report
No comments
Author Response
We are glad that with the revised version we have addressed all the concerns you have raised, and that our work could be now considered for publication in Sustainability.
Best regards,
The Authors
Reviewer 2 Report
The authors have made careful changes in response to the proposed revisions, although the following problems remain:
1. Originality: Although the article adds to the discussion of the relationship between green supply chains and digital supply chains, the authors should emphasize the innovative contributions of this paper. It is crucial to clearly state the specific contributions in both the introduction and conclusion sections, highlighting how this paper fills the existing research gap. By doing so, the authors can clearly articulate the unique insights and advancements provided by this study, thus distinguishing it from previous research. Therefore, it is recommended that the authors explicitly mention the contributions of this paper, their significance, and how they address the research gap in both the introduction and conclusion sections.
2. Research Framework: “The distribution of the articles in terms of journals shows a low level of concentration, with many different journals contributing to the publication of the 131 articles in our dataset......”. The number of publications on the subject in various journals should preferably be in the form of a graph, which will be more visual. By including a graph, readers can easily visualize the distribution of publications across different journals, making the information more accessible and facilitating a better understanding of the overall landscape. Therefore, it is recommended that the authors incorporate a graph to illustrate the number of publications in different journals, thereby enhancing the visual presentation and aiding in the comprehension of the data.
3. Conclusions: While the authors have included a concluding section, there is a need for a more comprehensive analysis of the degree of overlap between "green" and "digital" in specific areas such as big data, blockchain technology, the circular economy, transportation, and lean and agile approaches. By thoroughly examining the extent of overlap in these domains, the authors can provide a deeper understanding of the intersection between green practices and digital advancements. This analysis would shed light on the potential synergies, challenges, and opportunities that arise when combining green and digital concepts in each specific area. Therefore, it is recommended that the authors conduct a detailed analysis of the degree of overlapping between "green" and "digital" in the aforementioned domains to enrich the concluding section and provide valuable insights for future research and practice.
1. Quality of Communication: The manuscript contains formatting errors, specifically in lines 434-446 and 581-593, where the font size and paragraph spacing do not match the rest of the content. It is crucial to address these inconsistencies to ensure the overall uniformity and professionalism of the manuscript. The formatting should be carefully reviewed and adjusted accordingly in those specific sections to maintain a consistent style throughout the document. By rectifying these formatting errors, the manuscript will present a more polished and visually appealing appearance. Therefore, it is recommended that the authors thoroughly review the manuscript, paying particular attention to the mentioned lines, and make the necessary adjustments to ensure consistent font size and paragraph spacing throughout.
Author Response
Reviewer 2
Comments to the Author
Reviewer 2_1
The authors have made careful changes in response to the proposed revisions, although the following problems remain
Answer to Reviewer 2_1
We thank Reviewer 2 for acknowledging that we have properly addressed all the insightful comments provided to us. We hope that with this revised version we have finally handled the problems and criticisms you have highlighted to us.
Reviewer 2_2
Originality: Although the article adds to the discussion of the relationship between green supply chains and digital supply chains, the authors should emphasize the innovative contributions of this paper. It is crucial to clearly state the specific contributions in both the introduction and conclusion sections, highlighting how this paper fills the existing research gap. By doing so, the authors can clearly articulate the unique insights and advancements provided by this study, thus distinguishing it from previous research. Therefore, it is recommended that the authors explicitly mention the contributions of this paper, their significance, and how they address the research gap in both the introduction and conclusion sections.
Answer to Reviewer 2_2
We thank Reviewer 2 for this insightful comment. We have emphasized the contributions of our paper in the Introduction and Conclusion sections, highlighting the uniqueness of our results compared to previous studies.
Reviewer 2_3
Research Framework: “The distribution of the articles in terms of journals shows a low level of concentration, with many different journals contributing to the publication of the 131 articles in our dataset......”. The number of publications on the subject in various journals should preferably be in the form of a graph, which will be more visual. By including a graph, readers can easily visualize the distribution of publications across different journals, making the information more accessible and facilitating a better understanding of the overall landscape. Therefore, it is recommended that the authors incorporate a graph to illustrate the number of publications in different journals, thereby enhancing the visual presentation and aiding in the comprehension of the data.
Answer to Reviewer 2_3
We are grateful to Reviewer 2 for this suggestion. We have incorporated a bar chart identified as Figure 4 to better illustrate the distribution of publications among the most popular journals. In fact, given that the total number of journals that have published the 131 articles in our dataset is 58, a graph including all those 58 journals would have shown a limited visual clarity.
Reviewer 2_4
Conclusions: While the authors have included a concluding section, there is a need for a more comprehensive analysis of the degree of overlap between "green" and "digital" in specific areas such as big data, blockchain technology, the circular economy, transportation, and lean and agile approaches. By thoroughly examining the extent of overlap in these domains, the authors can provide a deeper understanding of the intersection between green practices and digital advancements. This analysis would shed light on the potential synergies, challenges, and opportunities that arise when combining green and digital concepts in each specific area. Therefore, it is recommended that the authors conduct a detailed analysis of the degree of overlapping between "green" and "digital" in the aforementioned domains to enrich the concluding section and provide valuable insights for future research and practice.
Answer to Reviewer 2_4
We are thankful to Reviewer 2 for the remarkable comment. We have revised the Conclusion section to deepen the analysis of our results and their implications for the business environment. In particular, we have better analyzed the reasons behind the different degrees of overlapping between the two domains, deepening the reflection about the cause behind such level of overlapping, potential benefits, synergies, and providing also suggestions for further exploration and development.
Reviewer 2_5
Quality of Communication: The manuscript contains formatting errors, specifically in lines 434-446 and 581-593, where the font size and paragraph spacing do not match the rest of the content. It is crucial to address these inconsistencies to ensure the overall uniformity and professionalism of the manuscript. The formatting should be carefully reviewed and adjusted accordingly in those specific sections to maintain a consistent style throughout the document. By rectifying these formatting errors, the manuscript will present a more polished and visually appealing appearance. Therefore, it is recommended that the authors thoroughly review the manuscript, paying particular attention to the mentioned lines, and make the necessary adjustments to ensure consistent font size and paragraph spacing throughout.
Answer to Reviewer 2_5
We are grateful to Reviewer 2 for pointing this out. We have amended the font size and paragraph spacing in lines 434-466 and 581-593. In addition, we have carefully revised the entire manuscript to make sure the font size, formatting and spacing would be uniform.
All the amendments implemented in the revised version of the paper have been highlighted in red. We hope that with this revised version we have addressed all the insightful concerns you have raised, and that our work could be now considered for publication in Sustainability.
Best regards,
The Authors
Round 3
Reviewer 2 Report
The authors have carefully revised the reviews.